# Transcription Factor TeMADS6 Coregulates Carotenoid Biosynthesis and Chlorophyll Degradation Resulting in Yellow-Green Petal Color of Marigold (*Tagetes erecta*)

**DOI:** 10.3390/plants14243763

**Published:** 2025-12-10

**Authors:** Chunling Zhang, Ke Zhu, Chujun Huang, Luan Ke, Yafeng Wen, Hang Li, Chaolong Yang, Zhengguo Tao, Yanhong He

**Affiliations:** 1Key Laboratory of Landscape Architecture of Jiangsu Province, College of Landscape Architecture, Nanjing Forestry University, Nanjing 210037, China; 2College of Landscape Architecture, Central South University of Forestry and Technology, Changsha 410004, China; 3National Key Laboratory for Germplasm Innovation & Utilization of Horticultural Crops, College of Horticulture and Forestry Sciences, Huazhong Agricultural University, Wuhan 430070, China; 4Guangzhou Leader Bio-Technology Co., Ltd., Guangzhou 510000, China

**Keywords:** carotenoids, chlorophyll, MADS, overexpression, marigold (*Tagetes erecta*)

## Abstract

Marigold (*Tagetes erecta*) is an important ornamental and industrial crop valued for its high lutein content. Although petal pigmentation during inflorescence development involves coordinated chlorophyll degradation and carotenoid biosynthesis, the transcriptional mechanisms regulating these processes remain poorly understood. Here, we identified a MADS-box transcription factor, TeMADS6, that coordinately regulates chlorophyll and carotenoid metabolism in marigold. Constitutive overexpression of *TeMADS6* resulted in yellow-green petals. HPLC analysis revealed that lycopene, antheraxanthin, violaxanthin, zeaxanthin, and lutein levels were substantially reduced in *TeMADS6*-overexpression lines, while chlorophyll content was significantly increased compared with wild-type plants. Transcriptome profiling revealed strong repressions of the carotenoid biosynthetic genes *TePSY1* and *TeHYDB* in transgenic florets. Moreover, the chlorophyll degradation gene *TeNYC1* and *TePPH2* were significantly downregulated, whereas *TeSGR2* was upregulated. Together, these findings demonstrate that *TeMADS6* acts as a dual-function transcriptional regulator controlling both chlorophyll degradation and carotenoid biosynthesis. This study provides new genetic resources for manipulating petal color and enhancing lutein accumulation in marigold, and advance understanding of the transcriptional networks orchestrating pigment metabolism during flower development.

## 1. Introduction

Chlorophylls and carotenoids, the two primary pigments in plant chloroplasts, determine the coloration of fruits and flowers [1,2]. Chlorophyll, a porphyrin-derived compound, serves as the primary photoreceptive pigment in photosynthesis, enabling the absorption and conversion of light energy [3]. Beyond its role in photosynthesis, chlorophyll exhibits antioxidant and antimutagenic properties [4]. Carotenoids are terpenoid compounds that function as accessory photosynthetic pigments and protect plants from oxidative stress by quenching reactive oxygen species [5]. Beyond their photosynthetic role, carotenoids contribute to flower and fruit coloration, thereby attracting pollinators and seed dispersers, which support plant reproduction [6]. Carotenoids are also important dietary nutrients for humans and animals, with antioxidant and immunomodulatory properties [6,7].

Genes involved in carotenoid biosynthesis have been well characterized in horticultural crops [1]. The pathway begins with the condensation of two geranylgeranyl pyrophosphate (GGPP) molecules by phytoene synthase (PSY), to produce colorless phytoene. Phytoene is subsequently converted to lycopene, the red-colored carotenoid, through desaturation and isomerization by phytoene desaturase (PDS), *ζ*-carotene desaturase (ZDS), ζ-carotene isomerase (ZISO), and carotenoid isomerase (CRTISO). Lycopene then undergoes cyclization by lycopene *β*-cyclase (LCYB) and lycopene *ε*-cyclase (LCYE), which splits the pathway into *α*- and *β*-branch derivatives. Subsequent modifications, including hydroxylation by *β*-ring carotene hydroxylase (HYDB) and carotene *ε*-ring carotene hydroxylase (HYDE), epoxidation by enzymes like zeaxanthin epoxidase (ZEP), lead to xanthophylls formation (lutein, zeaxanthin, antheraxanthin, neoxanthin, and violaxanthin). These xanthophylls are ultimately catabolized by carotenoid cleavage dioxygenases (CCDs) and 9-cis-epoxycarotenoid dioxygenase (NCED).

Chlorophyll biosynthesis and degradation pathways have been extensively studied [1]. Chlorophyll biosynthesis occurs within the chloroplasts, beginning with the conversion of glutamate to protoporphyrin IX. Magnesium chelatase (CHLI/CHLD/CHLH) inserts a magnesium ion into protoporphyrin IX to form Mg-protoporphyrin IX. This intermediate is methylated by the Mg-protoporphyrin IX methyltransferase (CHLM) to yield Mg-protoporphyrin IX monomethyl ester (MPE), which is converted to chlorophyllide a through reactions mediated by Mg-Proto IX monomethyl ester cyclase (CHL27) and NADPH-dependent protochlorophyllide oxidoreductase (PORA). Chlorophyll synthase (CS) catalyzes the addition of a phytol group to chlorophyllide *a* to from chlorophyll *a*, which can be further converted to chlorophyll *b* by chlorophyllide *a* oxygenase (CAO). Chlorophyll degradation begins with the reduction of chlorophyll *b* to chlorophyll *a*, catalyzed by chlorophyll *b* reductase (NOL/NYC). Chlorophyll *a* is then broken down by enzymes including STAY-GREEN (SGR), pheophytin pheophorbide hydrolase (PPH), and pheophorbide a monooxygenase (PAO), facilitating its dephytylation and subsequent catabolism into non-green metabolites.

The MADS-box transcription factor (TF) family is one of the largest families in plants and regulates various developmental processes, particularly floral organogenesis [8]. Recent studies have revealed that MADS-box proteins regulate carotenoid accumulation and thereby influence flower and fruit coloration [9,10,11]. In tomato (*Solanum lycopersicum*), MADS TFs such as SlCMB1, SlMADS1, and Agamous-like 1 (TAGL1) regulate carotenoid biosynthesis by modulating carotenogenic gene expression [12,13,14]. Similarly, the citrus (*Citrus sinensis*) and honeysuckle (*Lonicera japonica*) MADS proteins CsMADS6 and LjMADS36 promote carotenoid accumulation by directly activating carotenogenic gene transcription [9,10]. Some MADS TFs, such as citrus CsMADS3, coordinately regulate chlorophyll degradation and carotenoid biosynthesis [15]. Silencing *CaRIN* in pepper (*Capsicum annuum*) increases chlorophyll levels while decreasing carotenoid content [16]. Among these MADS TFs, FRUITFULL (FUL) proteins of the APETALA1 (AP1)/FUL subfamily regulate carotenoid biosynthesis. For instance, knockdown of tomato *SlFUL1* and *SlFUL2* decreases lycopene accumulation, resulting in yellow ripe fruits [17]. Similarly, citrus FUL protein CsMADS5 positively regulates carotenoid metabolism [18]. Despite their established roles in carotenoid metabolism, the potential involvement of *FUL* genes in chlorophyll synthesis has not yet been reported.

Marigold (*Tagetes erecta*), an annual herb of the Asteraceae, is highly valued for its ornamental traits, particularly its diverse petal colors ranging from white to deep orange-red. Marigold petals are rich in carotenoids, particularly lutein, which comprises approximately 90% of the total carotenoid. This makes marigold an important industrial source of natural lutein for the food, feed, chemical, and pharmaceutical industries. Marigold petal coloration is associated with carotenoid accumulation and chlorophyll degradation [19]. Darker petals accumulate higher carotenoid levels than lighter petals [20]. Moreover, carotenoid accumulation correlates with key carotenoid biosynthesis gene expression [21]. Carotenoid biosynthesis genes (*TePDS*, *TeZDS*, *TePSY*, and *TeLCYE*) are highly expressed in dark petals with high lutein content, while carotenoid cleavage genes are repressed [21,22,23]. Although carotenoid biosynthesis gene expression patterns have been well characterized in marigold, the transcriptional mechanisms regulating carotenoid biosynthesis remain poorly understood.

In this study, we identified a marigold MADS—box TF, *TeMADS6*, which was expressed more highly in the cream—white petals (cultivar ‘Vanilla’) than in orange petals (cultivar ‘Lady’). To investigate the function of *TeMADS6*, we overexpressed it in marigold. Functional analysis demonstrated that TeMADS6 regulated petal color by coordinating chlorophyll degradation and carotenoid biosynthesis through regulation of key genes in both pathways. These findings advance understanding of the transcriptional mechanisms controlling petal pigmentation in marigold and provide genetic resources for enhancing marigold ornamental and economic value.

## 2. Materials and Methods

### 2.1. Plant Materials

Two marigold cultivars were used: ‘Lady’ (deep-orange petals) from Chifeng Seno Horticulture Co., Ltd. (Chifeng, China) and ‘Vanilla’ (cream-white petals) from Thailand AmeriSeed Company (Chiang Mai, Thailand). Plants were grown in the experimental field at Central South University of Forestry and Technology (Changsha, Hunan Province, China; 113.01° E, 28.14° N). Inflorescence development in was classified into four key stages as described previously: S1 (Pre-flowering), characterized by tightly packed, green florets; S2 (Unopened), where the outermost florets elongated and began to develop pigmentation; S3 (Semi-opened), marked by the sequential elongation and pigmentation of florets from the outer to inner whorls; and S4 (Full bloom), during which all florets were fully expanded [24].

The yellow-petaled marigold cultivar ‘Milestone’, used in genetic transformation experiments, was obtained from Chifeng Seno Horticulture Co., Ltd. (Chifeng, China). Transgenic plants were grown in a growth chamber under a 16 h light/8 h dark photoperiod at 25 °C.

### 2.2. Total RNA Extraction, cDNA Synthsis, and RT-qPCR Analysis

Florets from both cultivars ‘Lady’ and ‘Vanilla’ were collected at four inflorescence developmental stages as previously described [24]. Total RNA was extracted using the RN33-PLANTpure Kit (Aidlab, Beijing, China). RNA integrity and concentration were assessed using a NanoDrop 2000 Spectrophotometer (Thermo Scientific, Waltham, MA, USA). cDNA was synthesized using the TRUEscript RT Kit (Aidlab, Beijing, China). Quantitative real-time PCR (qRT-PCR) was performed on an ABI7500 Light Cycle 96 system using the PerfectStart^®^ Uni RT&qPCR Kit (TransGen, Beijing, China). The 20 μL reaction mixture contained 2 μL cDNA, 0.4 μL of each primer, 10 μL of 2× PerfectStart Green qPCR SuperMix, and 7.2 μL double-distilled water. PCR cycling conditions were as follows: initial denaturation at 94 °C for 30 s, followed by 40 cycles of 94 °C for 5 s, 60 °C for 15 s, and 72 °C for 10 s. Each sample was tested in triplicate biological replicates. The *beta-actin* served as the endogenous reference. Relative gene expression was calculated using the 2^−ΔΔCt^ method. All the primers used for qRT-PCR experiments were listed in Appendix A.

### 2.3. Gene Isolation, Phylogenetic Tree Construction, and Multiple Sequence Alignment

Based on the marigold genomic data (GenBank assembly GCA_026213115.1), specific primers were designed to amplify the full-length coding sequence (CDS) of *TeMADS6* (TeORChr2g0068971) (Appendix A). MADS-box protein sequences from other plants species were downloaded from NCBI (https://www.ncbi.nlm.nih.gov/protein/, accessed on 13 May 2024) (Appendix A). A phylogenetic tree was constructed using MEGA 7.0 software with the neighbor-joining method and 1000 bootstrap replicates. Multiple sequence alignment was performed using Clustal X1.83 and visualized using Genedoc 2.7 software.

### 2.4. Marigold Transformation

The coding sequence of *TeMADS6* was amplified using primers with restriction sites (Appendix A) and inserted into the P2300 vector under the control of the CaMV *35S* promoter to generate the *35S:TeMADS6* overexpression construct. This recombinant vector was then transformed into *Agrobacterium tumefaciens* strain *EHA105* for marigold transformation using an established Agrobacterium-mediated method [25]. Briefly, sterile marigold leaves were immersed in a bacterial suspension (OD_600_ = 0.6–0.8) for 10 min. They were then transferred to a co-culture medium and incubated in the dark at 25 °C for 24 h. Following co-cultivation, the explants were transferred to a selection medium (MS + 1.0 mg/L 6-BA + 0.1 mg/L NAA + 50 mg/L kanamycin + 250 mg/L cefotaxime) to induce callus formation and shoot regeneration. Resistant shoots approximately 1 cm in height were excised and transferred to a rooting medium (1/2 MS + 0.5 mg/L IBA + 30 mg/L kanamycin + 250 mg/L cefotaxime). When the resulting transgenic plantlets had developed 5 to 6 leaves, they were transplanted into soil for further growth. Transgenic plants were further confirmed by genomic PCR using gene-specific primers (P2300-TeMADS6-F) and the universal primer 35S-F (Appendix A). The expression levels of *TeMADS6* in transgenic lines and wild-type ‘Milestone’ plants (WT) were analyzed by qRT-PCR. Petal color parameters of the outermost ray florets from transgenic and WT were measured using a CM-5 spectrophotometer (Konica Minolta, Japan). Three inflorescences were evaluated per line, with three petals analyzed per inflorescence.

### 2.5. Determination of Carotenoids and Chlorophyll Content

Florets at full bloom stage from transgenic lines (OE#1, OE#2, OE#3) and WT were collected and freeze-dried. Carotenoids were extracted and quantified by HPLC as previously described [26]. Carotenoids in marigold florets were quantified using an AB Sciex WTRAP 6500 LC-MS/MS system (MetWare, Wuhan, China), with concentrations determined from standard curves of authentic compounds. Experiments were performed in triplicate biological replicates. Peak areas for carotenoids (phytoene, *α*-carotene, *β*-carotene, and lycopene) and xanthophylls (lutein, violaxanthin, zeaxanthin, neoxanthin, and antheraxanthin) were quantified.

The extraction, detection, and quantification of total chlorophyll, chlorophyll *a*, and chlorophyll *b* in florets of *TeMADS6*-overexpression transgenic marigold lines (OE#1, OE#2, and OE#3) and WT were determined using acetone. Briefly, a 0.1 g sample was thoroughly homogenized using a tissue grinder under dim light conditions at −80 °C. The homogenate was then mixed with acetone and extracted for 3 h in complete darkness or while wrapped in aluminum foil. The resulting homogenate was then combined with acetone and incubated in the dark for 3 h, a process continued until the tissue residue became completely colorless. Finally, the absorbance of the acetone extract was measured at 645 nm and 663 nm using a microplate reader (EPOCH-SN, BioTek, Winooski, VT, USA). The experiments were performed with three biological replicates.

### 2.6. RNA-Seq and Data Analysis

The phenotypic divergence in floret color between transgenic and WT initially emerged at the semi-open stage of inflorescence (S3) (Appendix A). Furthermore, the significantly lower expression of *TeMADS6* in the lutein-rich cultivar ‘Lady’ than in the low-lutein cultivar ‘Vanilla’ at the S3 and S4 stages. Given that both the phenotype and key gene expression differences are manifested during these stages, we reasoned that S3 and S4 are critical for petal color determination in marigold. Therefore, to comprehensively investigate the molecular mechanisms underlying petal color alteration in *TeMADS6*-overexpression lines, we collected the florets at S3 and S4 stages inflorescences of the highest-expressing transgenic line (OE#1) and WT. These samples were subsequently used for total RNA extraction and transcriptome sequencing (RNA-seq). Three independent biological replicates were included for RNA-seq. Library construction was performed by Shanghai Majorbio Bio-pharm Technology Co., Ltd. (Shanghai, China), and sequencing was conducted using an Illumina HiSeq™ 2500 system (Illumina, San Diego, CA, USA). The sequencing data statistics were analyzed (Appendix A). Clean reads were mapped to the marigold reference genome (GCA_026213115.1) using HISAT and Bowtie2 software (http://tophat.cbcb.umd.edu/, accessed on 9 March 2025), with mapping ratios for each library recorded (Appendix A). The transcript length distribution was counted (Appendix A). The differentially expressed genes (DEGs) and Gene Functional Enrichment Analyses (GO and KEGG) were conducted on the online platform of the Majorbio Cloud Platform (www.majorbio.com). The gene expression levels were calculated using the number of fragments per kilobase of transcript sequence per million mapped reads (FPKM) method. The RNA-seq quality and read numbers were summarized in Appendix A. Raw data were submitted to the NCBI Short Read Archive (SRA) database under accession number PRJNA1330313.

### 2.7. Validation of RNA-Seq Data Using qRT-PCR

The DEGs related to carotenoid and chlorophyll metabolism were screened based on the criteria of a |log2 fold change| > 1, an adjusted *p*-value < 0.05, and an FPKM > 20. To validate RNA-seq data, these DEGs were quantified by qRT-PCR. The total RNA from florets at S3 and S4 stages from *TeMADS6*-overexpression transgenic lines and WT was extracted and reversed transcribed as described above. Gene-specific primers were designed for qRT-PCR validation. Each experiment was performed with three independent biological replicates.

### 2.8. Statistical Analysis

Statistical significance for data presented as means ± SD (n ≥ 3) was analyzed by One-way ANOVA
using SPSS 26. As determined by an independent samples t-tests, significance was denoted by asterisks (*p* < 0.01). Significance from Duncan’s test was indicated by different letters (*p* < 0.05 or *p* < 0.01).

## 3. Results

### 3.1. Bioinformatics Analysis and Expression Pattern of TeMADS6

In this study, we cloned *TeMADS6*, which encodes a protein of 214 amino acids. Multiple sequence alignment revealed that TeMADS6 is a typical MADS-box protein containing four highly conserved domains: the M-domain, I-region, K-box, and C-terminal domain (Figure 1a). Phylogenetic analysis revealed that TeMADS6 clustered in the FUL subclade and was most closely related to sunflower (*Helianthus annuus*) and chrysanthemum (*Chrysanthemum* × *morifolium*) FUL proteins (Figure 1b). Within this subclade, the FUL proteins from citrus (CsMADS5), bilberry (*Vaccinium myrtillus*) (TDR4), and tomato (SlFUL1, and SlFUL2) are reported to regulate carotenoid biosynthesis (Figure 1b). Expression pattern analysis revealed that *TeMADS6* expression levels decreased during inflorescence blooming and coloration in marigold cultivars ‘Lady’ and ‘Vanilla’ (Figure 1c). Its expression levels were consistently lower in the lutein-rich cultivar ‘Lady’ than in the low-lutein cultivar ‘Vanilla’, with a significant reduction observed particularly at the S3 and S4 stages (Figure 1c), suggesting that *TeMADS6* is involved in petal color and carotenoid biosynthesis.

### 3.2. Overexpression of TeMADS6 Alters Flower Coloration

To investigate *TeMADS6* function in marigold, we generated stable transgenic lines overexpressing *TeMADS6*. After antibiotic selection and PCR verification, five independents transgenic lines were obtained. Among them, three transgenic lines (OE#1, OE#2, and OE#3) exhibited a yellow-green petals compared with control plants (Figure 2a). QRT-PCR analysis further confirmed that *TeMADS6* transcript levels were significantly higher in these lines than in WT (Figure 2b).

To quantify petal color changes in transgenic plants, the color parameters *L**, *a**, and *b** of petals from transgenic lines (OE#1, OE#2, and OE#3) and WT were measured. Yellowness (*b**) and redness (*a**) values were significantly reduced in the *TeMADS6*-overexpression lines, while lightness (*L**) was not significantly different (Figure 2c). These results demonstrate that *TeMADS6* overexpression in marigold alters petal coloration.

### 3.3. TeMADS6 Regulates Carotenoid and Chlorophyll Accumulation in Marigold

To elucidate the biochemical basis for petal color alterations, we quantified carotenoid and chlorophyll in florets from transgenic lines (OE#1, OE#2, and OE#3) and WT. Consistent with the phenotypic changes, total carotenoid content was reduced by 17%, 11%, and 10% in OE#1, OE#2 and OE#3, respectively, compared with WT (Figure 3a). Specifically, the contents of lycopene, antheraxanthin, violaxanthin, zeaxanthin, and lutein were significantly reduced in the *TeMADS6*-overexpression lines than in WT (Figure 3b). Lutein content was reduced by 18%, 15%, and 15% in OE#1, OE#2 and OE#3, respectively, relative to WT. Neoxanthin content in OE#1 was significantly lower than in OE#2, OE#3 and WT (Figure 3b). Correspondingly, the *β*-carotene content was also significantly reduced in OE#1 relative to WT (Figure 3b). However, *α*-carotene levels did not differ significantly between transgenic and wild-type lines (Figure 3b).

In contrast to the carotenoid reductions, the total chlorophyll content in petals of *TeMADS6*-overexpressing marigold lines was significantly increased compared to WT (Figure 3c). Chlorophyll *b* content was significantly elevated in OE#1, while chlorophyll *b* content in OE#2 and OE#3 did not differ from WT (Figure 3d). Chlorophyll *a* level did not differ significantly between transgenic lines and WT (Figure 3d).

### 3.4. Transcriptome Sequencing and Gene Function Annotation

To investigate the molecular mechanisms underlying the petal color alterations in *TeMADS6*-overexpression lines, we performed transcriptome sequencing on florets at the semi-open (S3) and full-bloom (S4) stages of the highest-expressing transgenic line (OE#1) and WT. A total of 80.58 Gb of clean data was obtained, with a minimum of 6.71 Gb per sample. Cluster analysis and principal component analysis (PCA) confirmed high reproducibility among biological replicates (Pearson correlation coefficient > 0.9; Appendix A), indicating reliable RNA-seq data. Gene function annotation analysis revealed that a total of 27,672 unigenes were annotated. Of these, 23,653, 11,867, 25,533, 27,639, 22,633, and 23,658 unigenes were annotated in the GO, KEGG, EggNOG, NR, Swiss-Prot, and Pfam databases, respectively (Figure 4a and Appendix A). In particular, 151 unigenes were classified under ‘Secondary metabolites biosynthesis, transport, and catabolism’ in the EggNOG database (Figure 4b). GO annotation identified multiple unigenes involved in metabolic processes in OE_S3_vs_WT_S3 and OE_S4_vs_WT_S4 groups (Figure 4c). Furthermore, KEGG annotation identified ‘Metabolism’ as the most common pathway category, with more unigenes in OE_S3_vs_WT_S3 group than in OE_S4_vs_WT_S4 group (Figure 4d). These results suggest that overexpression of *TeMADS6* in marigold predominantly alters metabolite biosynthesis.

### 3.5. DEGs Identification and Functional Enrichment Analysis 

A total of 9808 DEGs were identified (Figure 5a). Among these, 2890 and 542 genes were upregulated in the OE_S3_vs_WT_S3 and OE_S4_vs_WT_S4 groups, respectively, while 1462 and 337 genes were downregulated in the OE_S3_vs_WT_S3 and OE_S4_vs_WT_S4 groups, respectively (Figure 5a). The total number of DEGs in OE_S3_vs_WT_S3 group was higher than in OE_S4_vs_WT_S4 group (Figure 5a). Venn diagram analysis showed that 1235 DEGs were shared between OE_S3_vs_WT_S3 and OE_S4_vs_WT_S4 groups (Figure 5b). KEGG enrichment analysis showed that the 1982 DEGs in OE_S3_vs_WT_S3 group were classified into three functional categories: organismal systems, environmental information processing, and metabolism (Figure 5c). In contrast to results in OE_S3_vs_WT_S3 group, the 4352 DEGs identified in OE_S4_vs_WT_S4 group were primarily classified into Environmental Information Processing and Metabolism. Further analysis revealed that DEGs in OE_S3_vs_WT_S3 and OE_S4_vs_WT_S4 groups were mainly enriched in metabolic pathways, such as phenylpropanoid biosynthesis, starch and sucrose metabolism, and photosynthesis (Figure 5d). Notably, pathways related to carotenoid biosynthesis were enriched in OE_S3_vs_WT_S3 group, but not in OE_S4_vs_WT_S4 group (Figure 5c,d), suggesting that carotenoid metabolite was primarily regulated at the S3 stage.

### 3.6. DEGs Involved in Carotenoid and Chlorophyll Metabolism Between TeMADS6-Overexpression and WT Plants

To elucidate the mechanism by which *TeMADS6* regulates carotenoid accumulation in marigold petals, a total of 14 genes involved in the carotenoid biosynthetic pathway were identified from transcriptome data (Figure 6a,b and Appendix A). DEGs analysis showed that two DEGs, *TePSY1* and *TeHYDB*, were differentially expressed in OE_S3_vs_WT_S3 or OE_S4_vs_WT_S4 (Figure 6a and Appendix A). *TePSY1* expression in *TeMADS6*-overexpression line (OE#1) was significantly lower than that in WT. Similarly, *TeHYDB* expression was lower in *TeMADS6*-overexpressing line than in WT at the S3 stage (Figure 6a). To validate the RNA-seq data, these two DEGs were quantified by qRT-PCR (Figure 6c). Correlation analysis revealed a strong correlation between the qRT-PCR and RNA-seq for *Te**PSY1* (R^2^ > 0.80, *p* < 0.05). Conversely, *TeHYD**B* showed weaker correlation (R^2^ < 0.60, *p* > 0.05) (Figure 6c).

Furthermore, we identified 18 genes involved in chlorophyll metabolism (Figure 7a,b and Appendix A). Among these, three genes (*TeNYC1*, *TeSGR2*, and *TePPH2*) encoding key chlorophyll metabolism enzymes were differentially expressed in OE_S3_vs_WT_S3 or OE_S4_vs_WT_S4 (Figure 7a and Appendix A). Specifically, at the S3 stage, *TePPH2* expression was significantly lower in the *TeMADS6*-overexpressing line than in WT (Figure 7a). At the S4 stage, the expression of *TeNYC1* was significantly reduced in the *TeMADS6*- overexpressing line. Conversely, *TeSGR2* was significantly upregulated (Figure 7a). Correlation analysis showed strong correlation between qRT-PCR and RNA-Seq for *TeNYC1* and *TeSGR2* (R^2^ > 0.85, *p* < 0.05). However, *TePPH2* showed weaker correlation (R^2^ < 0.70, *p* > 0.05) (Figure 7c).

### 3.7. Differentially Expressed TFs Under Overexpression TeMADS6 in Marigold

RNA-Seq revealed 350 and 150 differentially expressed TFs in OE_S3_vs_WT_S3 and OE_S4_vs_WT_S4 groups, respectively. Of these, 285 and 76 TFs were up-regulated, whereas 65 and 64 TFs were down-regulated in OE_S3_vs_WT_S3 and OE_S4_vs_WT_S4 groups, respectively (Appendix A). TFs from the ERF, NAC, MYB, bHLH, HB, and WRKY families represented the majority of differentially expressed TFs (Figure 8a,b and Appendix A). ERF and bHLH TFs represented the highest number of differentially expressed TFs in OE_S3_vs_WT_S3 and OE_S4_vs_WT_S4 groups, respectively (Figure 8a,b and Appendix A).

## 4. Discussion

Carotenoids and chlorophyll are major pigments in plants, exhibiting both synergistic and antagonistic relationships that influence the coloration of petals and fruits. However, the transcriptional regulation of carotenoid and chlorophyll metabolism, particularly in horticultural ornamental plants, remains poorly understood. In this study, *TeMADS6* was found to regulate petal color alteration in marigold
likely
by modulating the expression of genes involved in chlorophyll degradation and carotenoid biosynthesis.

### 4.1. TeMADS6 Regulates Carotenoid Degradation and Chlorophyll Biosynthesis

MADS-box TFs are involved in diverse biological processes, including plant development, stress responses, and metabolic regulation [10,27,28]. Many MADS-box genes have been identified in horticultural species, such as tomato, citrus, and kiwifruit (*Actinidia deliciosa*), where they play critical roles in fruit ripening and carotenoid biosynthesis [13,15,29]. In this study, we characterized TeMADS6, a FUL-clade MADS-box protein that clusters with other carotenoid biosynthesis regulators, such as CsMADS5 from citrus and SlFUL1 from tomato (Figure 1a,b). *TeMADS6* expression was higher in creamy-white petals than in deep-orange petals (Figure 1c). Previous studies have demonstrated that lutein content in dark-colored marigold petals is higher than in light-colored petals [20,30]. These results suggest that *TeMADS**6* may be involved in regulating carotenoid biosynthesis and petal coloration.

Overexpression of *TeMADS6* in marigold resulted in a yellow-green petal phenotype, with decreased carotenoid levels, particularly xanthophylls, and increased chlorophyll content (Figure 2a,c, and Figure 3). These findings suggest that *TeMADS6* regulates both carotenoid and chlorophyll biosynthesis. This is consistent with previous reports in citrus, where the MADS-box gene *CsMADS3* promotes carotenoid biosynthesis and chlorophyll degradation [15]. However, in this study, *TeMADS6* inhibits carotenoid biosynthesis while promoting chlorophyll accumulation in marigold petals. Furthermore, a recent study in pepper (*Capsicum annuum*) reveals that the MADS-box gene *CaRIN* represses the chlorophyll *a*/*b*-binding protein gene *CaLhcb-P4*, increasing chlorophyll content and reducing carotenoid accumulation [16]. These findings demonstrate the diverse role of *MADS* genes in regulating carotenoid and chlorophyll metabolism.

### 4.2. TeMADS6 Induces Expression of Key Chlorophyll and Carotenoid Metabolic Genes

In plants, carotenoids biosynthesis and chlorophyll degradation are complex regulatory processes. These metabolic pathways are associated with changes in endogenous gene expression. In this study, functional annotation analysis revealed that ‘metabolic process’ was the predominant category in the KEGG annotation, with 151 unigenes in ‘Secondary metabolites biosynthesis, transport and catabolism’ (Figure 4b,d). GO annotation analysis further demonstrated that numerous unigenes were classified as ‘metabolic process’ (Figure 4c), indicating broad involvement of these genes in various metabolic pathways.

The enzymes NYC, PPH, and SGR play important roles in the regulation of chlorophyll metabolism [31,32]. NYC promotes leaf senescence by catalyzing the conversion of chlorophyll *b* to chlorophyll *a* [32]. Suppression of *NYC* results in a stay-green phenotype [33,34]. Conversely, overexpression of *NYC* promotes chlorophyll degradation [35]. Likewise, *Arabidopsis PPH* mutant retains chlorophyll during senescence and remains green [36]. In this study, the transcript levels of *TeNYC1* and *TePPH2* were significantly decreased in *TeMADS6*-overexpression lines, which may explain the increased chlorophyll content in transgenic plant petals. Among chlorophyll catabolic enzymes, SGR is a critical enzyme in the degradation pathway of green plants, catalyzing the first step of chlorophyll degradation by removing magnesium from chlorophyll *a*. In *Arabidopsis*, *SGR2* inhibits chlorophyll degradation in senescing leaves [37]. Overexpression of *SGR2* results in a stay-green phenotype and delayed leaf senescence, while knockout of *sgr2* leads to premature leaf yellowing [37]. Similarly, apple *MdSGR2* negatively regulates chlorophyll degradation [38]. Additionally, silencing of *SlSGR1* in tomato results in elevated lycopene and *β*-carotene accumulation [39]. Tomato fruit ripening is characterized by carotenoid accumulation and a chlorophyll degradation [40]. In the present study, overexpression of *TeMADS6* in marigold upregulated *TeSGR2* expression (Figure 7a,c), consistent with the increased chlorophyll content in transgenic plant petals (Figure 3c,d). Additionally, although the functions of *NYC*, *PPH*, and *SGR* are well characterized in plants, their upstream regulatory mechanisms remain unclear. Our findings suggest that *TeMADS6* may act as an upstream regulator of these genes, a hypothesis requiring further validation.

PSY is the first rate-limiting enzyme in carotenoid biosynthesis, catalyzing the production of phytoene from two molecules of geranylgeranyl pyrophosphate. In horticultural crops such as tomato [41], apple [42], and citrus [43], *PSY* promotes lycopene accumulation in fruits. *PSY* also regulates flower coloration in ornamental plants. For instance, the white petals of California poppies (*Eschscholzia californica*) result from *EcPSY* mutations that reduce carotenoid content [43]. Furthermore, silencing of *OgPSY* in *Oncidium* orchid petals leads to decreased lutein (total xanthophyll) levels, resulting in a new orchid variety with white petals [44]. In this study, we found that overexpression of *TeMADS6* significantly downregulated *TePSY1* expression, decreasing carotenoid accumulation, including lycopene, lutein, and zeaxanthin (Figure 3a,b and Figure 6a,c). Correspondingly, the expression of *TeHYDB* was also significantly lower in the *TeMADS6*-overexpressing transgenic lines compared to the WT (Figure 6a). This observation is consistent with findings in other species. For example, in kiwifruit, mutation of the *HYDB*-like gene *AcBCH1* significantly reduces the contents of zeaxanthin and lutein [45], whereas overexpression of *CsBCH2* in citrus callus results in an elevated xanthophyll proportion [46]. Taken together, these results suggest that diminished carotenoid accumulation is likely attributable to the downregulation of both *TePSY1* and *TeHYDB*.

In addition to MADS transcription factors, other TF families have also been demonstrated to participate in the regulation of these two metabolic biosynthesis pathways. For example, overexpression of kiwifruit *AdMYB7D* in tobacco enhances carotenoid accumulation while reducing chlorophyll content in leaves [47]. Suppression of the *SlMYB72* gene in tomato promotes chlorophyll accumulation while suppressing lycopene biosynthesis [48]. Similarly, silencing the NAC transcription factor *NOR*-like1 in tomato decreases ethylene synthesis, delays fruit softening and chlorophyll degradation, as well as diminishes lycopene accumulation [49]. In citrus plants, ERF, MYB, WRKY, and bHLH TFs have also been reported to participate in regulating the biosynthesis of both chlorophyll and carotenoids [50,51,52,53]. For example, knockdown of *CrWRKY* expression markedly increases chlorophyll content and reduces carotenoid accumulation, consequently causing the fruit peel to retain a pale green color during development [50]. In this study, overexpression of *TeMADS6* in marigold significantly altered the expression of a wide array of TFs (Figure 8), which in turn may regulate chlorophyll and carotenoid metabolism in the transgenic lines.

### 4.3. Regulatory Model of TeMADS6 and Its Importance in Marigold Petal Color

Previous studies have revealed that MADS-box transcription factors regulate carotenoid and chlorophyll biosynthesis by directly binding to the promoters of carotenogenic genes to control their expression [10,13,15]. Based on our findings, we propose a potential model in which *TeMADS6* coordinately regulates carotenoid and chlorophyll metabolism in marigold petals (Figure 9). *TeMADS6* overexpression likely suppresses carotenoid biosynthesis through downregulation of *TePSY1*, while simultaneously inhibiting chlorophyll degradation by repressing *TeNYC1* and activating *TeSGR2*.

Consistent with our findings, citrus CsMADS3 has been shown to coordinate chlorophyll degradation and carotenoid biosynthesis [15]. However, unlike the results, CsMADS3 mainly promotes chlorophyll degradation and carotenoid accumulation by activating the expression of the *CsSGR*, *CsLCYB*, and *CsPSY1* genes, indicating that the *MADS* genes function within a complex regulatory network controlling carotenoid and chlorophyll synthesis. In this study, *TeMADS6* overexpression in marigold resulted in reduced *TePSY1* expression (Figure 6c), leading to decreased carotenoid accumulation. In citrus, the FUL protein CsMADS5 promotes carotenoid accumulation by positively regulating the expressions of the *LCYB*, *PDS*, and *PSY* genes [18]. Similarly, a recent study in chili pepper found that the *MADS* gene *CaRIN* mainly represses the expression of the chlorophyll *a*/*b* binding protein CaLhcb-P4, thereby increasing chlorophyll content and reducing carotenoid accumulation [16], demonstrating that *MADS* genes play different roles in the molecular regulation of carotenoid accumulation in plants.

Chlorophyll degradation and carotenoid biosynthesis are critical processes for fruit ripening and petal coloration in many horticultural plants. Our study revealed that *TeMADS6* overexpression led to yellow-green petals through coordinated regulation of key genes in both chlorophyll degradation and carotenoid biosynthesis. Similarly, the brown tomato varieties involve both chlorophyll degradation and carotenoid biosynthesis, a trait controlled by the expression levels of genes from these two metabolic pathways [54]. Likewise, citrus fruit ripening is accompanied by chlorophyll breakdown and carotenoid accumulation [15,50]. Furthermore, distinct expression patterns of genes involved in chlorophyll and carotenoid metabolic pathways have been observed in florets of cauliflower with varying color phenotypes [55]. Therefore, understanding the transcriptional coordination between chlorophyll degradation and carotenoid biosynthesis is essential for manipulating fruit ripening and flower coloration. However, research on the transcriptional regulatory mechanisms that coordinately control carotenoid and chlorophyll metabolism in plants, particularly in ornamental flowers, remains limited. Therefore, TeMADS6 represents a promising candidate for modifying petal color and enhancing lutein content in marigold petals, with potential applications in other horticultural crops. To elucidate the precise regulatory mechanisms underlying chlorophyll and carotenoid metabolism, further studies are needed.

## 5. Conclusions

In summary, our study demonstrates that TeMADS6 is a key TF regulating carotenoid biosynthesis and chlorophyll degradation in marigold petals. *TeMADS6* overexpression resulted in a yellow-green petal characterized by decreased carotenoid accumulation and increased chlorophyll levels. Through transcriptome analysis, we identified key genes in these metabolic pathways that were differentially expressed in *TeMADS6*-overexpressing marigold lines. These findings provide novel insights into the transcriptional coordination of pigment metabolism in ornamental plants and offer a genetic basis for improving petal color and carotenoid accumulation in marigold.

## Figures and Tables

**Figure 1 plants-14-03763-f001:**
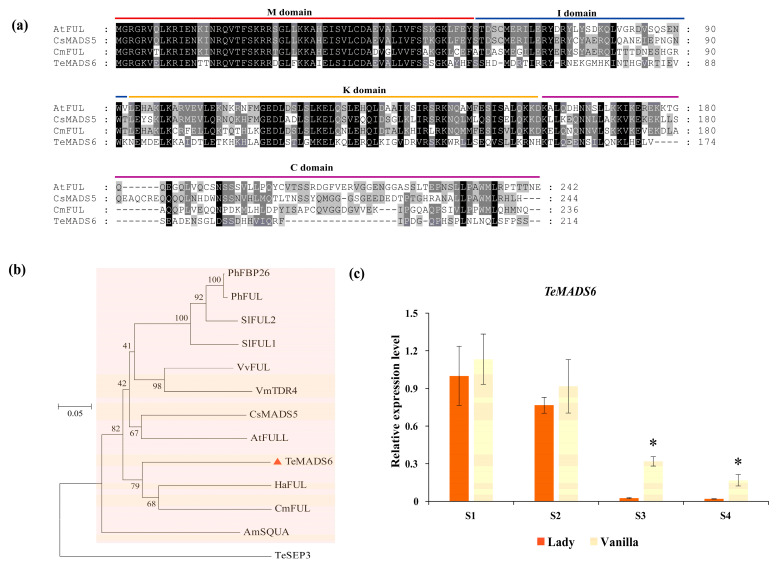
Bioinformatics analysis and expression pattern of *TeMADS6*. (**a**) Multiple sequence alignment of TeMADS6 and its homologs from other plant species. The four conserved domains of the MADS proteins were marked with different-colored horizontal lines. (**b**) Phylogenetic analysis of TeMADS6 and related proteins from other plant species. The TeMADS6 was marked with a red triangle. The proteins marked with an orange box belonged to the SQUA/FUL subfamily. The scale bar represents 0.05 substitutions per site. Marigold TeSEP3 was used as the outgroup. (**c**) The expression patterns of *TeMADS6* at different inflorescence development stages of ‘Lady’ with deep-orange petals and ‘Vanilla’ with cream-white petals. S1: the inflorescence at pre-flowering stage; S2: the inflorescence at unopened inflorescence stage; S3: the inflorescence at semi-open stage; S4: the inflorescence at full-bloom stage; Significant differences between ‘Lady’ and ‘Vanilla’ at the same inflorescence developmental stage were analyzed using independent samples *t*-tests (* *p* < 0.01). Statistical significance for data presented as means ± SD (n = 3) was analyzed by One-way ANOVA in SPSS 26.

**Figure 2 plants-14-03763-f002:**
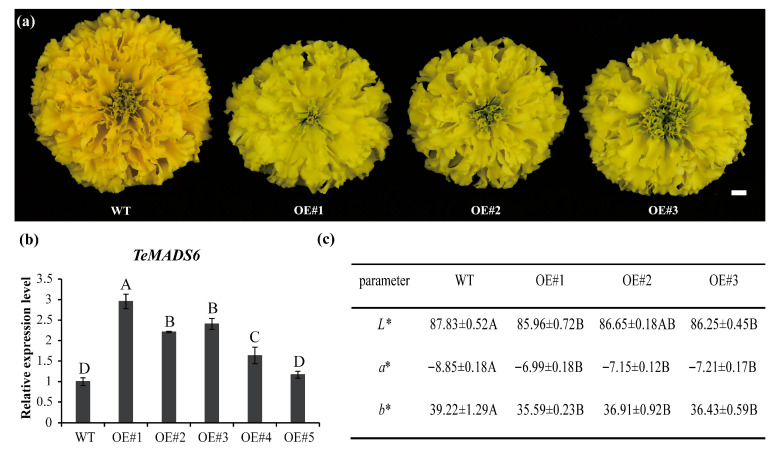
Phenotypic analysis of transgenic marigold with overexpression of *TeMADS6*. (**a**) Phenotypic changes of *TeMADS6*-overexpression transgenic marigold lines and wild-type (WT) control. (**b**) *TeMADS6* expression levels in overexpression transgenic marigold lines and WT. Data represent means ± SD (n = 3 biological replicates). Different letters indicate statistically significant differences (Duncan’s test, *p* < 0.01). (**c**) Statistics of petal color coefficients from *TeMADS6*-overexpression transgenic marigold lines and WT. WT: wild-type ‘Milestone’ plants; OE#1: OE-*TeMADS6*#1, OE#2: OE-*TeMADS6*#2, and OE#3: OE-*TeMADS6*#3. The data were presented as means ± SD (n = 3 biological replicates, each with 3 technical replicates). Different letters indicate statistically significant differences (Duncan’s test, *p* < 0.01).

**Figure 3 plants-14-03763-f003:**
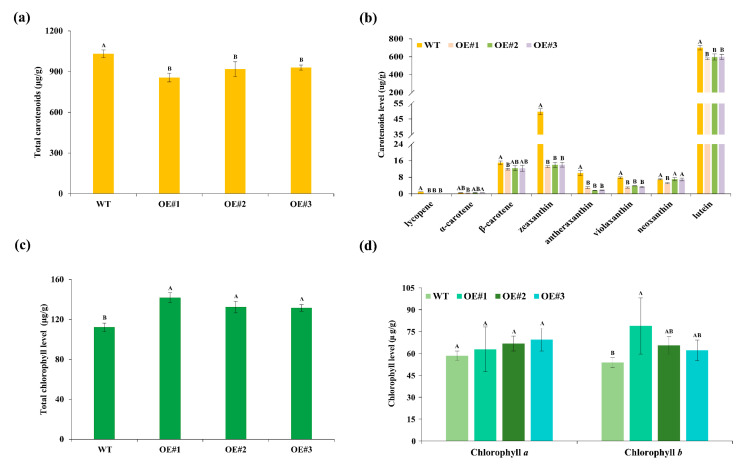
Carotenoid and chlorophyll content in floret of *TeMADS6*-overexpression lines and WT. (**a**) Total carotenoid content; (**b**) Individual carotenoid contents. (**c**) Total chlorophyll content; (**d**) Chlorophyll *a* and *b* content. WT: wild-type ‘Milestone’ plants, OE#1: OE-*TeMADS6*#1, OE#2: OE-*TeMADS6*#2, and OE#3: OE-*TeMADS6*#3. The data were presented as means ± SD (n = 3 biological replicates). Different letters indicate statistically significant differences (Duncan’s test, *p* < 0.01).

**Figure 4 plants-14-03763-f004:**
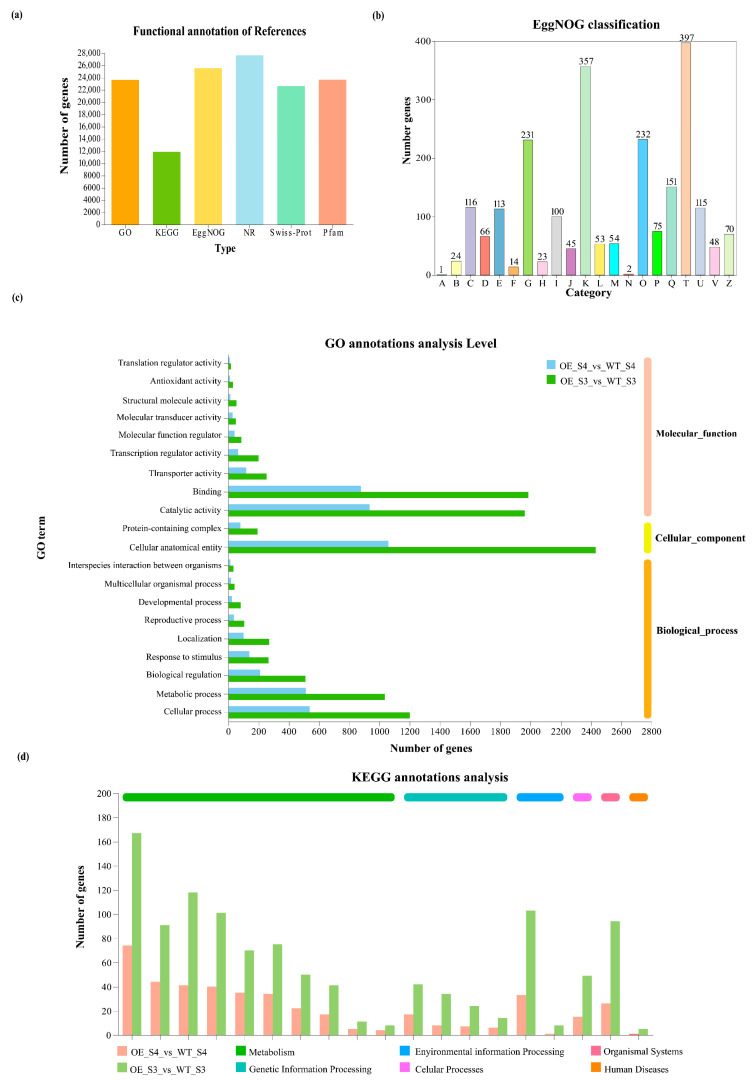
Functional annotation of unigenes from RNA-seq. (**a**) Statistics of unigenes annotated in different databases; (**b**) Functional classification of marigold unigenes in EggNOG categories. A: RNA processing and modification; B: Chromatin structure and dynamics; C: Energy production and conversion; D: Cell cycle control, cell division, chromosome partitioning; E: Amino acid transport and metabolism; F: Nucleotide transport and metabolism; G: Carbohydrate transport and metabolism; H: Coenzyme transport and metabolism; I: Lipid transport and metabolism; J: Translation, ribosomal structure and biogenesis; K: Transcription; L: Replication, recombination and repair; M: Cell wall membrane/envelope biogenesis; N: Cell motility; O: Posttranslational modification, protein turnover, chaperones; P: Inorganic ion transport and metabolism; Q: Secondary metabolites biosynthesis, transport and catabolism; T: Signal transduction mechanisms; U: Intracellular trafficking, secretion, and vesicular transport; W: Defense mechanisms; Z: Cytoskeleton; (**c**) Functional classification of marigold unigenes in GO categories. WT: wild-type ‘Milestone’ plants, OE: OE-*TeMADS6*#1; (**d**) Functional classification of marigold unigenes in KEGG categories.

**Figure 5 plants-14-03763-f005:**
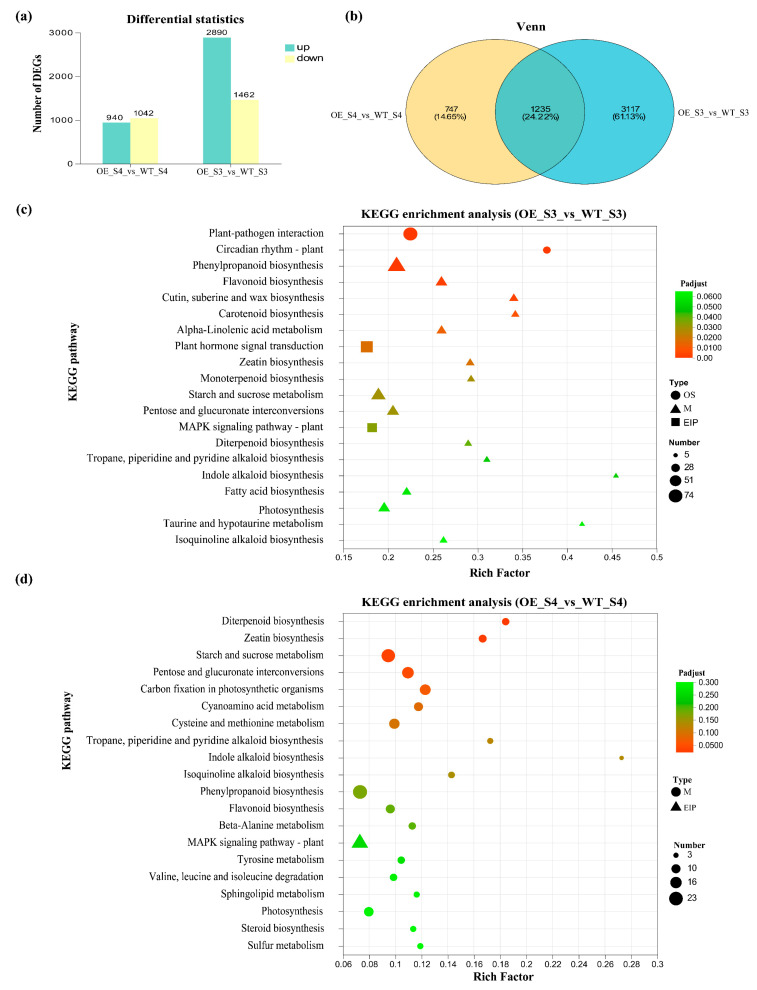
Analysis of DEGs in OE_S3_vs_WT_S3 and OE_S4_vs_WT_S4. (**a**) Numbers of upregulated and downregulated DEGs; (**b**) Venn diagram of DEGs in OE_S3_vs_WT_S3 and OE_S4_vs_WT_S4; (**c**) KEGG enrichment analysis of DEGs in OE_S3_vs_WT_S3; (**d**) KEGG enrichment analysis of DEGs in OE_S4_vs_WT_S4. WT: wild-type ‘Milestone’ plants, OE: OE-*TeMADS6*#1.

**Figure 6 plants-14-03763-f006:**
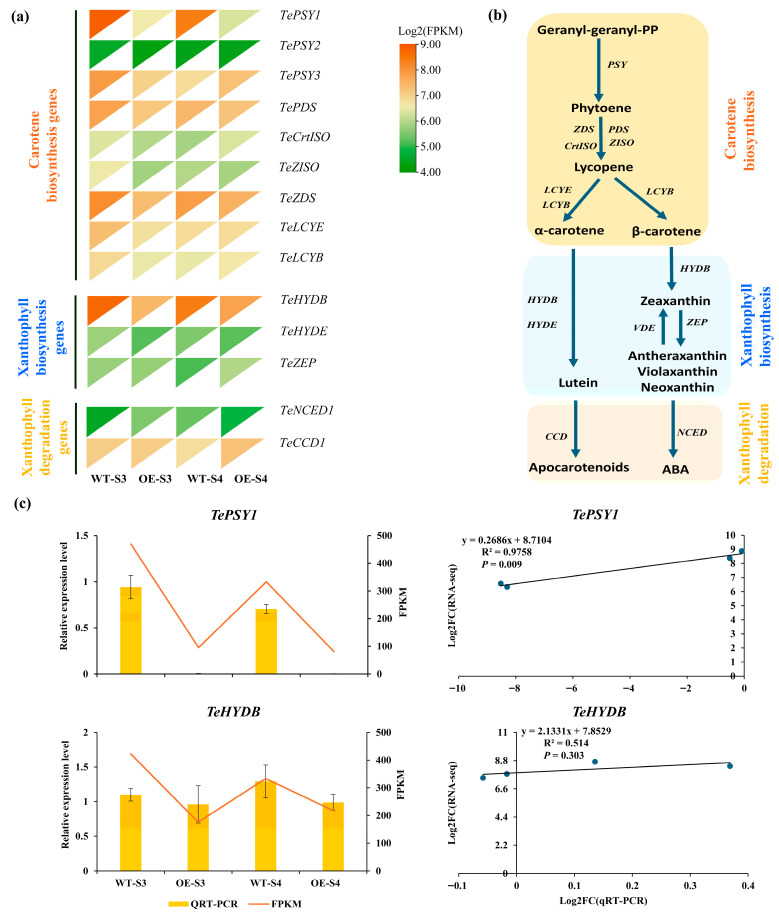
The expression levels of DEGs involved in carotenoid biosynthesis in transgenic lines and WT. (**a**) Expression Heatmap of carotenoid biosynthesis-related genes from OE_S3_vs_WT_S3 and OE_S4_vs_WT_S4. (**b**) Metabolic pathways involved in biosynthesis of carotenoids. (**c**) Validation of RNA-seq date by qRT-PCR. The data were presented as means ± SD of three biological replicates and three technical replicates. WT: wild-type ‘Milestone’ plants, OE: OE-*TeMADS6*#1.

**Figure 7 plants-14-03763-f007:**
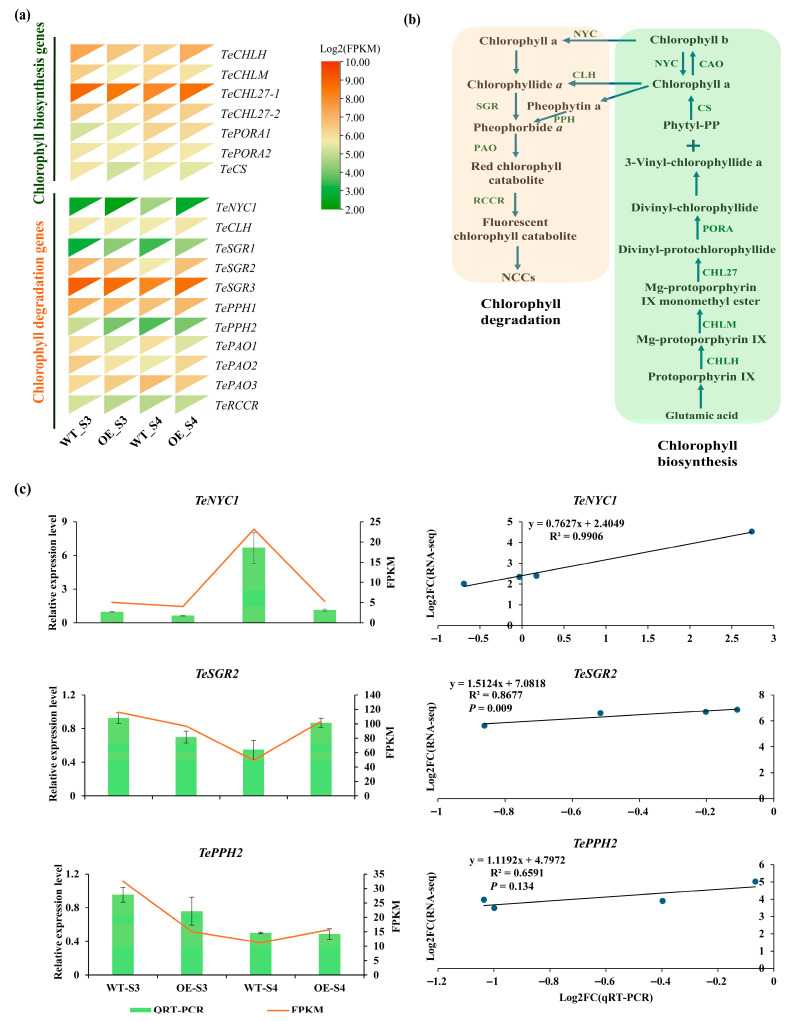
Expression of DEGs involved in chlorophyll metabolism in *TeMADS6*-overexpression lines and WT. (**a**) Expression Heatmap of chlorophyll metabolism-related genes from OE_S3_vs_WT_S3 and OE_S4_vs_WT_S4. (**b**) The chlorophyll metabolism pathway. (**c**) Validation of RNA-seq date by qRT-PCR. The data were presented as means ± SD of three biological replicates and three technical replicates. WT: wild-type ‘Milestone’ plants, OE: OE-*TeMADS6*#1.

**Figure 8 plants-14-03763-f008:**
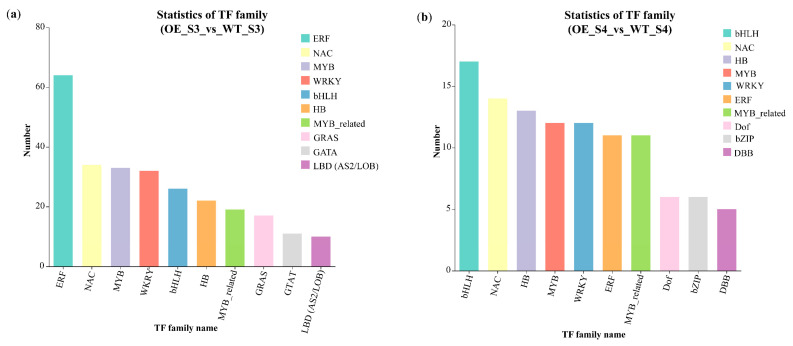
Differentially expressed TF families in OE_S3_vs_WT_S3 and OE_S4_vs_WT_S4 groups. (**a**) Numbers of differentially expressed TFs by family in OE_S3_vs_WT_S3; (**b**) Numbers of differentially expressed TFs by family in OE_S4_vs_WT_S4. WT: wild-type ‘Milestone’ plants, OE: OE-*TeMADS6*#1.

**Figure 9 plants-14-03763-f009:**
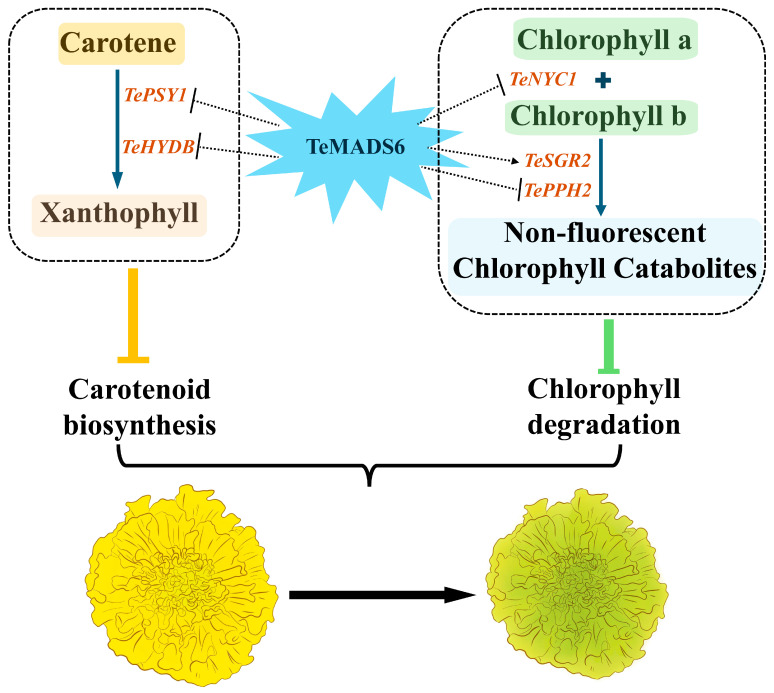
Proposed model illustrating how *TeMADS6* regulates chlorophylls, and carotenoids metabolism in marigold petals. *TeMADS6* overexpression reduces carotenoid biosynthesis by downregulating *TePSY1* and *TeHYDB*, and inhibits chlorophyll degradation by suppressing *TeNYC1* and *TePPH2* while promoting *TeSGR2* expression. Thick blue arrows in the dotted box indicate the simplified reaction flows in the pathway of carotenoids and chlorophyll. The dashed or dashed inhibitory arrows represent that *TeMADS6* overexpression either promotes or inhibits the expression of associated genes. The direct regulation by TeMADS6 (indicated with dashed lines) remains to be determined. The orange and green inhibitory arrows indicate that the overexpression of *TeMADS6* in marigold inhibits carotenoid accumulation and chlorophyll degradation, respectively. The thick black arrow denotes that *TeMADS6* overexpression causes the petal color to transition from yellow to yellow-green.

## Data Availability

The original contributions presented in this study are included in the article/Appendix A. Further inquiries can be directed to the corresponding author.

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
