# Peer review of "Transcription Factor TeMADS6 Coregulates Carotenoid Biosynthesis and Chlorophyll Degradation Resulting in Yellow-Green Petal Color of Marigold (*Tagetes erecta*)"

_plants, 2025, doi:10.3390/plants14243763_

Round 1

Reviewer 1 Report (Previous Reviewer 1)

Comments and Suggestions for Authors
  1. Grammar and spelling mistakes need to be corrected. For example, line 201, abstract, ect…….
  2. The authors found that the expression of TeMADS6 is significantly lower in the lutein-rich cultivar 'Lady' than in the low-lutein cultivar 'Vanilla at the S3 and S4 stages. Why performed transcriptome sequencing on florets from the semi-open (S1) and full-bloom (S2) stages.
  3. The KEGG analysis should focus on the DEGs between S1 and S2. Why the authors only showed the DEGS in S1 or S2.
  4. The authors demonstrated the transcription factor TeMADS6 regulates Carotenoid 2 Biosynthesis and Chlorophyll Degradation in Marigold through RNA-seq. However, the RNA-seq analysis cannot fully support the conclusion. The authors should thoroughly analyze RNA-seq data.
  5. Why only select the analysis of TePSY1, TeHYDB, TeNYC1, TeSGR2, and TePPH1. The other genes?The TeMADS6 can directly regulate the TePSY1, TeHYDB, TeNYC1, TeSGR2, and TePPH1. The authors should identify downstream target genes to support the conclusion.

Author Response

  1. Grammar and spelling mistakes need to be corrected. For example, line 201, abstract, ect…….

Response: We sincerely thank the reviewer for pointing out the need for grammatical and spelling corrections. We have thoroughly reviewed the entire manuscript, line by line, to address these issues. We have engaged a professional editing service to thoroughly revise the entire manuscript. We believe the manuscript is now significantly improved in terms of language quality.

  1. The authors found that the expression of TeMADS6 is significantly lower in the lutein-rich cultivar 'Lady' than in the low-lutein cultivar 'Vanilla at the S3 and S4 stages. Why performed transcriptome sequencing on florets from the semi-open (S1) and full-bloom (S2) stages.

Response: We apologize for the confusion caused by the inconsistent sample naming in our manuscript. In this study, the samples from four different inflorescence developments used for the qRT-PCR experiments were collected from the marigold cultivars 'Vanilla' and 'Lady'. These four different inflorescence developmental stages were designated as St1 (Pre-flowering), St2 (Unopened), St3 (Semi-opened), and St4 (Full bloom). In contrast, the samples used for transcriptome sequencing were obtained from the marigold 'Milestone' cultivar. To distinguish samples from different marigold cultivars, we named these two inflorescence developmental stages used for RNA-seq as S1 (Semi-opened) and S2 (Full bloom). To clarify, the samples S1 and S2 used for RNA-seq actually correspond to the same inflorescence developmental stages as the qRT-PCR samples St3 and St4, respectively. To prevent any potential misunderstanding, we have accordingly renamed the samples used for transcriptome sequencing to S3 and S4. This revision has been consistently applied to all relevant locations in the text and figures, please see the revised manuscript for detail  (Line 190,192, 212, 299, 309, 311, 330-345, 347-350, 356-357, 359, 370-371, 366-367, 372-373, 383, 387-389, 393, 395-397 Figure 4-8, Figure S1-S2, and Table S7-S12).

  1. The KEGG analysis should focus on the DEGs between S1 and S2. Why the authors only showed the DEGS in S1 or S2.

Response: Thank you for raising this point. We appreciate the opportunity to clarify our analytical focus. The primary objective of RNA-seq was to identify genes differentially expressed between the transgenic lines and the wild-type plants, not directly between the developmental stages S3 and S4 themselves. Therefore, our KEGG analysis was intentionally performed on the sets of DEGs identified in the S3 and S4 samples separately, each set representing the comparison of transgenic lines vs wild-type plants at that specific stage. This approach allows us to pinpoint the biological processes and pathways that are perturbed by the transgene at each distinct developmental point.

  1. The authors demonstrated the transcription factor TeMADS6 regulates Carotenoid 2 Biosynthesis and Chlorophyll Degradation in Marigold through RNA-seq. However, the RNA-seq analysis cannot fully support the conclusion. The authors should thoroughly analyze RNA-seq data.

Response: We thank the reviewer for this insightful comment regarding the need for a more thorough analysis of the RNA-seq data. We agree that a deeper dive into the transcriptomic data would strengthen our conclusions.

As suggested, we have now performed a comprehensive re-analysis of DEGs, with a specific focus on TF families. This analysis revealed that TeMADS6 not only regulates genes directly involved in carotenoid biosynthesis and chlorophyll degradation but also influences the expression of numerous other TFs such as MYB, bHLH, NAC, ERF families (Figure 8), which are known to play roles in pigment metabolism and plant development.

We have integrated these new findings into the results section (Line 390-401, Figure 8) and expanded our discussion to interpret the potential role of these TFs in the observed metabolic changes (Line 478-492). We believe this additional layer of analysis suggests that TeMADS6 might act as a master regulator orchestrating a complex transcriptional network.

  1. Why only select the analysis of TePSY1, TeHYDB, TeNYC1, TeSGR2, and TePPH1. The other genes?The TeMADS6 can directly regulate the TePSY1, TeHYDB, TeNYC1, TeSGR2, and TePPH1. The authors should identify downstream target genes to support the conclusion.

Response: We thank you for this important question. In this study, the candidate genes TePSY1, TeHYDB, TeNYC1, TeSGR2, and TePPH1 were selected based on our DEG criteria (|log2FC|>1, P-adjust < 0.05, FPKM > 20). Other genes involved in these two metabolic pathways were excluded as they did not meet these thresholds. We fully acknowledge that the regulatory relationships between TeMADS6 and genes such as TePSY1, TeNYC1, and TeSGR2 remain speculative. This hypothesis is grounded in prior evidence that MADS-box transcription factors can directly control carotenoid and chlorophyll metabolic genes in other plants (Zhang et al., 2018; Wang et al., 2024; Zhu et al., 2023) (line 494-496). For example, in citrus, MADS3, MADS5, and CrMADS6 control carotenoid accumulation by directly regulating carotenogenic genes. For a more accurate representation of their potential regulatory relationships, we had already used dashed lines in the schematic diagram to indicate these putative, indirect, or unverified interactions (Figure 9). We agree that direct validation is crucial. As suggested, we plan to use Y1H and EMSA assays in future studies to definitively identify the direct downstream targets of TeMADS6.

Reviewer 2 Report (Previous Reviewer 2)

Comments and Suggestions for Authors

The submitted manuscript may be recommended for publication in the journal in the presented form.

Author Response

We would like to express our sincere gratitude for your time and effort in reviewing our manuscript. We truly appreciate your valuable contribution to the review process.

We were pleased to be informed that you have no further comments on our revised manuscript. Your insightful feedback during the initial round of review was immensely helpful in improving the quality of our work. Thank you once again for your dedication and for helping us enhance our paper.

Round 2

Reviewer 1 Report (Previous Reviewer 1)

Comments and Suggestions for Authors

The authors has improved the manuscript. I agree to accept the manuscript in the current format.

Author Response

We would like to express our sincere gratitude for your time and effort in reviewing our manuscript. We truly appreciate your valuable contribution to the review process.

We were pleased to be informed that you have no further comments on our revised manuscript. Your insightful feedback during the initial round of review was immensely helpful in improving the quality of our work. Thank you once again for your dedication and for helping us enhance our paper.

This manuscript is a resubmission of an earlier submission. The following is a list of the peer review reports and author responses from that submission.

Round 1

Reviewer 1 Report

Comments and Suggestions for Authors
  1. The sentences in the text still require refinement to enable readers to better grasp their meaning. For example, “For example, citrus CsMADS3 promotes fruit ripening by simultaneously enhancing the accumulation of both pigments via direct transcriptional regulation [18]” as well as other places.
  2. In the Introduction section, the content of “Previous studies have indicated that the process of color change in marigold petals is accompanied by the accumulation of carotenoids and the degradation of chlorophyll [21]”, it would be preferable if more specific examples could be provided.
  3. Explain the rationale for selecting the TeMADS6 gene for study. Although the author provided a brief response to my query, it has not resolved my confusion. Given the numerous members within the MADS transcription factor family, why did the author select TeMADS6 for investigation?
  4. line 172, as genetically modified material, is the marigold in question the yellow-flowered variety or the cultivar “Lady” with deep-orange petals or ‘Vanilla’ with cream-white petals?
  5. Experimental control: The description of the “wild-type” plants used as controls is vague. What exactly are marigolds and wild marigolds referring to?
  6. 250 , 418, 419 lines, genes should be italicized. Check whether the formatting of proper nouns in the article is correct.
  7. Line 383, 402, OE_vs_WT, ??
  8. The author appears to have not yet resolved the issue of abbreviations within the manuscript. Please refer to the author guidelines for comprehensive revisions. At line 312, WT: wild-type plants. Where wild-type plants has been previously defined in the text, it may be abbreviated as WT.
  9. Why were WT and OE strains from the S1 and S2 periods selected for transcriptomic analysis?
  10. Is the Venn diagram in Figure 5b correct?
  11. Is the horizontal axis of TePSY1 and TeHYDB in Figure 6c correct?
  12. All images should be clear, concise and straightforward, enabling readers to grasp their meaning readily. For instance, in Figures 6a and 7a, are the transcription levels of these genes in the pathway increased or suppressed?
  13. Figure 8 lacks a clear mechanism explanation. The authors indicate that TeMADS6 reduces carotenoid biosynthesis by downregulating TePSY1 expression, while simultaneously inhibiting chlorophyll degradation by suppressing TeNYC1 expression and promoting TeSGR2 In the carotenoid biosynthesis pathway and chlorophyll degradation pathway, does TeMADS6 influence carotenoid synthesis and chlorophyll degradation solely by regulating the expression of TePSY1, TeNYC1, and TeSGR2?
  14. Please amend the reference format in your manuscript in accordance with the author guidelines.
  15. Supplementary Materials 7 and 8 should present the results in the tables in a more concise and comprehensible manner, for instance by indicating whether adjustments are significantly upward or significantly downward.

Comments on the Quality of English Language

The sentences in the text still require refinement to enable readers to better grasp their meaning.

Reviewer 2 Report

Comments and Suggestions for Authors

The manuscript under consideration is devoted to the study of the role of the transcription factor TeMADS6 in the regulation of flower color in marigolds using the example of two varieties, Lady and Vanilla, differing in color, as well as on several transgenic plants with superexpression of the gene of the studied transcription factor, obtained during Agrobacterium-mediated transformation of the Milestone variety. The expert has one very significant remark to the authors of this manuscript related to the lack of an evidence base for the 5 transgenic plants presented in the work. The reference [28 - Yu, X.; Wang, Y.; Liu, Y.; Yi, Q.; Chen, W.; Zhu, Y.; Duan, F.; Zhang, L.; He, Y. Establishment of Agrobacterium tumefaciens-mediated Genetic Transformation System of Marigold (Tagetes erecta). The publication year [Journal of Integrative Plant Biology 2023, 58, 760-769, doi:10.1007/s11240-005-9031-9] on the cited journal's website does not correspond to the year of publication, as volume 58 contains publications from 2016 and the cited manuscript is not found on the pages indicated by the authors. It is unclear how these transgenic plants were obtained, how the selection was carried out, there is no confirmation of the transgenic status of the resulting plants, etc. Given that this part of the work is the core upon which the entire work is built, this manuscript in its present form cannot be recommended for publication.